# Isotoxic High-Dose Stereotactic Body Radiotherapy (iHD-SBRT) Versus Conventional Chemoradiotherapy for Localized Pancreatic Cancer: A Single Cancer Center Evaluation

**DOI:** 10.3390/cancers14235730

**Published:** 2022-11-22

**Authors:** Martin Manderlier, Julie Navez, Matthieu Hein, Jean-Luc Engelholm, Jean Closset, Maria Antonietta Bali, Dirk Van Gestel, Luigi Moretti, Jean-Luc Van Laethem, Christelle Bouchart

**Affiliations:** 1Department of Radiation Oncology, HUB Institut Jules Bordet, Université Libre de Bruxelles, Rue Meylenmeersch 90, 1070 Brussels, Belgium; 2Department of Radiation Oncology, CHU de Charleroi, Boulevard Zoé Drion 1, 6000 Charleroi, Belgium; 3Department of Hepato-Biliary-Pancreatic Surgery, Hôpital Universitaire Erasme, Université Libre de Bruxelles, 1070 Brussels, Belgium; 4Faculty of Medicine, Université Libre de Bruxelles, 1070 Brussels, Belgium; 5Department of Radiology, Institut Jules Bordet, Université Libre de Bruxelles, 1070 Brussels, Belgium; 6Department of Radiology, Hopitaux Iris Sud, 1190 Brussels, Belgium; 7Department of Gastroenterology, Hepatology and Digestive Oncology, HUB Bordet Erasme Hospital, Université Libre de Bruxelles, Route de Lennik 808, 1070 Brussels, Belgium

**Keywords:** radiotherapy, pancreatic cancer, stereotactic radiotherapy, neoadjuvant therapy, chemoradiotherapy

## Abstract

**Simple Summary:**

Pancreatic ductal adenocarcinoma (PDAC) is an aggressive tumour associated with poor prognosis. The only potentially curative treatment is an oncological surgical resection. To increase the probability of resection, the use of neoadjuvant treatment is explored and can include chemoradiotherapy (CRT) or stereotactic body radiotherapy (SBRT). Given the lack of direct comparison between the two modalities, we retrospectively compared the clinical outcomes of patients treated for localized PDAC by isotoxic high dose SBRT (iHD-SBRT) with those of patients treated with conventional CRT in the same cancer center. The oncological outcomes showed that iHD-SBRT seems to be a promising option and may offer an improvement in overall survival in comparison to conventional CRT for localized PDAC. Further investigations are required to identify the exact role of SBRT and the optimal therapeutic neoadjuvant sequence.

**Abstract:**

Given the lack of direct comparative evidence, we aimed to compare the oncological outcomes of localized pancreatic ductal adenocarcinoma (PDAC) treated in the same tertiary cancer center with isotoxic high-dose stereotactic body radiotherapy (iHD-SBRT) or conventional chemoradiotherapy (CRT). Biopsy-proven borderline/locally advanced patients treated with iHD-SBRT (35 Gy in 5 fractions with a simultaneous integrated boost up to 53 Gy) or CRT (45–60 Gy in 25–30 fractions) were retrospectively included from January 2006 to January 2021. The median overall survival (mOS) was evaluated trough uni- and multivariate analyses. The progression free survival (PFS) and the 1-year local control (1-yLC) were also reported. Eighty-two patients were included. The median follow-up was 19.7 months. The mOS was in favour of the iHD-SBRT group (22.5 vs. 15.9 months, *p* < 0.001), including after multivariate analysis (HR 0.39 [CI95% 0.18–0.83], *p* = *0*.014). The median PFS and the 1-yLC were also significantly better for iHD-SBRT (median PFS: 16.7 vs. 11.5 months, *p* = 0.011; 1-yLC: 75.8 vs. 39.3%, *p* = 0.004). In conclusion, iHD-SBRT is a promising RT option and may offer an improvement in OS in comparison to CRT for localized PDAC. Further validation is required to confirm the exact role of iHD-SBRT and the optimal therapeutic sequence for the treatment of localized PDAC.

## 1. Introduction

Pancreatic ductal adenocarcinoma (PDAC) is a highly aggressive solid tumour associated with poor prognosis mainly due to the lack of symptoms, leading to a delayed diagnosis, and the high frequency of distant metastases [1]. With nearly 1700 deaths a year in Belgium and 44,000 in Europe, PDAC ranks fourth in the cancer-related mortality classification. It is estimated that by 2030, PDAC will reach the second place in the Western world [2,3]. All stages combined, the overall survival (OS) rate at 5 years is only 7%, an oncologic surgical resection being the only potentially curative treatment [1,4]. Localized PDACs are classified into resectable (up to 15% of the cases), borderline resectable (BR, around 15% of the cases) and locally advanced (LA, around 25%) according to the tumour’s relationship with the surrounding major vessels and the related probability to obtain a microscopically complete (R0) resection [5]. In order to increase this probability and better select the patient for surgery, the use of neoadjuvant therapies (chemotherapy and/or radiation therapy [RT]) has been explored and the exact neoadjuvant therapeutic sequence still needs to be validated. For a majority of LA tumours, oncological surgery will not be deemed feasible, hence RT may also be used as a definitive treatment option [4,5]. Considerable debates took place regarding the optimal RT scheme and the target volume to use for both neoadjuvant and definitive PDAC treatments. In particular, the stereotactic body radiation therapy (SBRT) technique is a promising technique which was recently explored in several observational and phase II trials, including ours [6,7]. SBRT offers multiple advantages compared to conventional long-course chemoradiation therapy (CRT) such as the shorter duration of treatment (1 week versus 4–6 weeks), the possibility of delivering higher biologically effective dose (BED) safely to the tumour, and its easy incorporation into a full neoadjuvant sequence [6]. However, as recently illustrated by the randomized phase II ALLIANCE A021501 trial, low to moderate-BED_10_ SBRT is not a solution as reflected in the disappointing clinical results obtained for the SBRT arm in this trial [8]. It is now known that the delivery of a BED_10_ > 60–70 Gy seems to be a predictor of prolonged survival in several studies [6,9,10,11,12]. In order to deliver high BED_10_ to the tumour without impairing the safety of the critical closest gastrointestinal organs at risk (OARs), we integrated an innovative isotoxic high-dose SBRT (iHD-SBRT) into a total neoadjuvant treatment sequence [7] and the same iHD-SBRT sequence was also used as a definitive treatment strategy for “never resectable” LA patients. As our group and others recently reported, modern techniques to safely deliver high-BED_10_ SBRT with conventional or magnetic resonance (MR) Linac showed promising clinical results but still require to be further validated [6,7,13,14]. Sparse data are available in the literature regarding the direct comparison between conventional CRT and SBRT and, to our knowledge, none for high-BED_10_ SBRT. Therefore, in order to provide more information about the optimal treatment strategy for localized PDAC, our aim was to compare the clinical outcomes of patients treated for localized PDAC by iHD-SBRT with those of patients treated with conventional CRT in the same tertiary cancer center.

## 2. Materials and Methods

### 2.1. Patient Selection

This study was approved by the Institutional Review Board of Institut Jules Bordet under the approval number CE3285. We retrospectively included all the patients with the following criteria: biopsy-proven BR or LA adenocarcinoma, age ≥ 18 years, no evidence of metastatic disease at baseline or after induction chemotherapy, treated with iHD-SBRT or conventional CRT between January 2006 and January 2021 at our institution. Patients treated for a local recurrence after prior surgery or RT were excluded.

BR and LA resection status were defined according to the NCCN criteria [5]. For the iHD-cohort, the resectability status had previously been prospectively assessed by a centralized multidisciplinary oncological board (MOC) including dedicated pancreatic surgeons and radiologists. For the conventional CRT cohort, the baseline abdominal computed tomography (CT)-scan and/or the magnetic resonance imaging (MRI) available at diagnostic were all retrospectively reviewed by an experienced pancreatic radiation oncologist (CB) to correctly establish the resectability status with the same criteria. In case of doubt, a dedicated gastrointestinal radiologist (JLE) reviewed the cases.

### 2.2. Radiation Therapy and Chemotherapy

From January 2006 until December 2017, BR/LA PDAC patients were treated with conventional CRT (25–30 fractions, total dose: 45–60 Gy with concomitant chemotherapy). The choice of the type and length of induction chemotherapy was left to the local oncologist and the decision to perform a surgical exploration or not after CRT was evaluated during a MOC. All patients underwent CT simulation with intravenous contrast in supine position with arms above the head. Over the years, various RT devices for movement reduction were used (cradle, vacuum bag, etc.). A Clinical Target Volume (CTV) was created using an expansion of approximately 1 cm from the gross tumour volume (GTV) which was then further enlarged to ensure coverage of the elective nodal regions around the superior mesenteric vessels, portal vein and celiac axis. The PTV was generated using a 0.5 to 1.0 cm expansion from the CTV. Conventional RT was delivered using three-dimensional conformal radiation therapy (3D-RT) for the oldest treatments, afterwards by intensity-modulated RT (IMRT) and volumetric modulated arc therapy (VMAT). The main dose constraints for OARs were as follows: for kidneys, D_mean_ < 18 Gy; for spinal cord, D_max_ < 45 Gy; for stomach and for small bowel, V45 < 195 cc, and for liver, D_mean_ < 30 Gy.

Since January 2018, all the localized PDAC patients were treated according to our total neoadjuvant treatment (TNT) strategy which included (1) an induction by modified FOLFIRINOX (mFFX: fluorouracil, irinotecan and oxaliplatin) or gemcitabine plus nab-Paclitaxel (Gem/nP; in case of intolerance or no response to mFFX after an intermediate restaging at 2 to 4 cycles) for ideally 6 cycles (a minimal number of 3 cycles was required), (2) treatment by iHD-SBRT (isotoxic dose prescription [IDP] with a D_max (0_._5 cc)_ < 53 Gy in 5 fractions) without concomitant chemotherapy and (3) a surgical exploration in case of no progression after a full restaging 4 to 7 weeks after the completion of iHD-SBRT. In case the tumour was deemed “never resectable” at our centralized MOC or the patient was inoperable, the iHD-SBRT was used as a definitive treatment and no further treatment was administered until progression. We previously described in detail the iHD-SBRT treatment as well as the fiducials markers insertion [7,15]. Briefly, prior to the contrast-enhanced CT scan, a four-dimensional (4D)-CT scan was performed to assess respiratory motion. The use of an abdominal compression belt (ZiFix^TM^, QFix, Avondale, PA, USA) was required in case of fiducial respiratory motion >5 mm in any direction, hence a new 4D-CT, with belt, was also performed. A specialized radiologist systematically reviewed the contouring of the GTV. No CTV was delineated as elective nodes were not included in the treatment volume. A tumour vessels interface (TVI) structure was created by including the whole circumference of major abdominal vessels in direct contact with the GTV. An internal target volume (ITV), accounting for respiratory motion and based on all the CT scan sequences available, was created for both GTV and TVI. PTV1 encompassed the ITVs plus a 3 mm margin. PTV2 was created by subtracting the sum of critical gastrointestinal OAR planning organ at risk volumes (PRVs) from the PTV1. PTV3 was an expansion of 3 mm from ITV_TVI_. An IDP was applied and therefore the dose prescription was not based on the target volume but on OAR tolerance levels [16]. For this, OAR-based RT following dose constraints were applied: for PRV stomach, duodenum, colon and small bowel, D_max (0_._5 cc)_ < 35 Gy, V_30 Gy_ < 2 cc; for PRV spinal cord, V_20 Gy_ < 1 cc, and for kidneys, D_mean_ < 10 Gy and V_12 Gy_ < 25%. The delivered dose was individually adapted and maximized to the highest achievable level in PTV2 and particularly in PTV3. iHD-SBRT was delivered using VMAT plans designed by the Monaco^TM^ planning system via a Monte Carlo algorithm. Details about the surgery, adjuvant chemotherapy and follow-up have been previously described [7].

### 2.3. Patient Characteristics

The baseline characteristics of the two cohorts are described in Table 1. The CRT and iHD-SBRT group had comparable median age, gender ratio, tumour diameter, pretreatment serum levels of CA19.9, stage and resection status. For the whole cohort, the median age was 60.2 (IQR 53.0–67.7) and the median diameter of the tumour was 37.5 mm (IQR 32.0–45.0). The median pretreatment serum level of CA19.9 was 86.4 kU/l (IQR 14.3–502.0) and the rates of BR and LA tumours were, respectively, 43.9 and 56.1%. The CRT group had significantly more tumours located in the body or tail than the iHD-SBRT group (39.0 versus 17.1%, *p* = 0.03). Induction chemotherapy consisted of mFFX or Gem-Np for 29.3% of the CRT patients and for all of the iHD-SBRT cohort (mFFX, a secondary shift to Gem/nP had to be performed in less than 10% of the patients). For the rest of the CRT cohort, the induction chemotherapy consisted of gemcitabine alone in 26.8%, gemcitabine combined with another agent (nab-paclitaxel, cisplatine, 5FU, etc.) in 29.3%; 24.4% had no induction chemotherapy. The median number of chemotherapy cycles was 3 (IQR 0–5) in the CRT group and 7 (IQR 6–8) in the iHD-SBRT group (*p* < 0.001). The median duration of induction chemotherapy was 2.1 months (IQR 0.8–3.3) for CRT and 3.7 months (IQR 2.6–4.6) for iHD-SBRT (*p* < 0.001). After RT, an oncological resection was performed in 9.8% and 46.3% of the cases for the CRT and iHD-SBRT groups, respectively (*p* < 0.001).

### 2.4. Clinical Outcomes

The median overall survival (mOS) was the primary endpoint. Median progression free survival (mPFS) and local control (LC) were also evaluated.

OS was calculated from the date of diagnosis to the date of last follow-up or death from any cause. PFS was calculated from the date of diagnosis to the date of the first loco-regional and/or distant metastatic progression or death from any cause. The 1-year LC was calculated from the end date of the RT treatment to the date of the first loco-regional failure. Loco-regional failure was defined as any progression meeting the RECIST criteria [17] for the irradiated PDAC and the loco-regional lymph nodes or any LRR after oncological surgery.

### 2.5. Statistical Analyses

Statistical analyses were performed using Stata 14. The normal distribution of the data was verified using histograms, boxplots, and quantile–quantile plots whereas the equality of variances was checked using the Levene test.

Categorical data were described by percentages and numbers whereas continuous variables were described by their median and P25–P75. Since most continuous data followed an asymmetric distribution, we decided to use non-parametric tests for all these variables (Wilcoxon test) in order to highlight significant differences between the medians (P25–P75) observed in the different groups of patients. Regarding categorical data, Chi² tests were used for the different analyses. Finally, survival functions were plotted using the Kaplan–Meier method and compared by log-rank test.

Univariate Cox regression models were used to study the mortality risk associated with RT treatments and the potential confounding factors. In multivariate Cox regression models, the mortality risk associated with RT treatments was only adjusted for significant confounding factors during univariate analyses. These different confounding factors were introduced hierarchically in the different multivariate Cox regression models.

Proportional hazard assumptions were assessed by statistical tests and graphical diagnostics based on the scaled Schoenfeld residuals to verify the validity of the final model.

The results were considered significant when the *p*-value was < 0.05.

## 3. Results

### 3.1. Patient Characteristics

In total, 82 patients were eligible and included in the clinical outcome analysis (RCT *n* = 41; iHD-SBRT *n* = 41). Patient characteristics are described in detail in Section 2.3 and Table 1.

### 3.2. Radiotherapy Treatments Characteristics

The PTV characteristics and related BED10 are described in detail in Table 2. The median PTV1 volume of iHD-SBRT was significantly lower than the volume of the conventional CRT’s PTV (99.6 vs. 422.7 cc, *p* < 0.001). The related BED10 Dmean and Dmax of the GTVs and global PTVs were also significantly higher for iHD-SBRT (Dmean BED10 GTV: 71.7 vs. 64.8 Gy [*p* = 0.002]; Dmean BED10 PTV: 66.1 vs. 60.3 Gy [*p* < 0.001]; Dmax BED10 PTV: 106.1 vs. 68.4 Gy [*p* < 0.001]).

### 3.3. Oncological Outcomes

The median follow-up was 19.7 months (IQR 14.8–24.0). The mOS was 15.9 months (IC95% 14.7–19.6) and 22.5 months (IC95% 20.5–26.5) for CRT and for iHD-SBRT, respectively (*p* < 0.001) (Figure 1). The 2-year OS rates were also in favour of the iHD-SBRT group (10.0% vs. 43.9%; *p* = 0.001). The mPFS from diagnosis was, respectively, 11.5 months (IC95% 8.4–14.1) and 16.7 months (IC95% 10.0–19.5) in favour of the iHD-SBRT (*p* = 0.011). A trend was shown for the loco-regional PFS (17.4 vs. 21.7 months [*p* = 0.060]) and the 1-year LC was in favour of the iHD-SBRT group (39.3 vs. 75.8, *p* = 0.004). The distant metastatic PFS was not statistically different (13.6 vs. 17.5 months [*p* = 0.09]). The resection rates after CRT and iHD-SBRT were, respectively, 9.8 and 46.3%, and for R0 resection rates (at 0 mm)—33.3 and 73.7%. OS and PFS were also analysed in the subgroup of 59 patients without oncological resection, 37 in the CRT group and 22 in the iHD-SBRT group. The mOS was also in favour of the iHD-SBRT group (15.9 vs. 20.7 months; *p* = 0.048). (Figure 2). There was no significant difference in median PFS between the iHD-SBRT group and the CRT group (11.3 vs. 8.0 months; *p* = 0.931).

Univariate Cox regression analyses for mortality risk in PDAC patients treated with CRT and iHD-SBRT were performed and are described in detail in Table 3. The following factors were significantly associated with a lower mortality risk: the number of induction chemotherapy cycles (4–8: Hazard ratio [HR] 0.47 [IC95% 0.27–0.81] and >8: HR 0.23 [IC95% 0.09–0.54], *p* = 0.001), the chemotherapy induction duration (≥2 & <4 months: HR 0.46 [IC95% 0.26–0.81] and ≥4 months: HR 0.38 [IC95% 0.20–0.72], *p* = 0.005), an induction with modern multi-agent chemotherapy (mFFX or Gem/Np) (HR 0.42 [IC95% 0.20–0.84], *p* = 0.025), oncological resection (HR 0.47 [IC95% 0.28–0.81], *p* = 0.009), and treatment with iHD-SBRT (HR 0.39 [IC95% 0.24–0.64], *p* < 0.001). Multivariate Cox regression analyses for mortality risk associated with RT treatments in patients with localized PDAC were performed and are detailed in the Table 4. After adjusting for the main significant confounding factors highlighted during univariate analyses, multivariate Cox regression analyses demonstrated that unlike conventional CRT, iHD-SBRT was significantly associated with a lower mortality risk in patients with localized PDAC (HR 0.39 [CI95% 0.18–0.83], *p* = 0.014).

## 4. Discussion

In the current study, our aim was to compare the clinical outcomes of our localized PDAC patients homogeneously treated with a TNT approach including iHD-SBRT with those of our patients treated with conventional CRT in the same tertiary cancer center. For localized PDAC, the role of RT in the neoadjuvant or definitive setting remains a subject of controversy and still need to be validated in randomized controlled trials. Since the phase III LAP07 trial for LA cancers failed to prove a survival benefit over chemotherapy alone, conventional CRT was gradually relegated to a secondary role [18]. However, the trial presented several limitations such as the use of gemcitabine alone as induction chemotherapy and a poor RT quality assurance [6,18]. Therefore, the results of the phase III CONKO-007 trial in which LA patients received an induction chemotherapy, mainly by mFFX, followed or not by CRT (50.4 Gy in 28 fractions with concomitant gemcitabine) were awaited. Due to delayed patient accrual, the primary endpoint was shifted from OS to R0 resection rate (RR); the first results were presented recently. With 525 patients included, the R0 RR did not show a significant difference (30 vs. 42% for CRT, *p* = 0.143) except for the circumferential resection margin (CRM)-R0 RR (15 vs. 33% for CRT, *p* = 0.001) and there was no statistically significant benefit for OS and PFS [19]. Altered types of CRT had also been studied, such as in the phase III PREOPANC-1 trial in which 248 resectable and BR patients were randomized between immediate surgery versus neoadjuvant hypofractionated CRT (36 Gy in 15 fractions; with concomitant gemcitabine). The CRT arm failed to improve PFS and OS; however, the long-term results recently published after a median follow-up of 59 months showed a moderate improvement in survival in favour of the neoadjuvant CRT (15.7 vs. 14.3 months, *p* = 0.025) [20,21]. In parallel, the SBRT technique allowing for the delivery of (nearly) ablative doses to the tumour in few sessions (one to five) was also studied. In the randomized phase II Alliance A021501, BR patients received an induction with mFFX +/− SBRT (33 Gy in five fractions with simultaneous integrated boost (SIB) up to 40 Gy at TVI or 25 Gy in five fractions) [8]. The primary endpoint, 18-month OS rate, was, respectively, 67.9 vs. 47.3% in disfavor of the SBRT arm. However, the results of this study should be considered with great caution as the trial was suspended for futility after the R0 RR interim analyses failed for the SBRT arm and thus was widely underpowered for SBRT. Of the 55/67 patients allocated to the SBRT arm, only 40 finally received SBRT and 12.5% of them received the palliative RT scheme of 25 Gy in five fractions [8]. In addition, the type of RT used in the Alliance trial correspond to a low/moderate-BED_10_ (37.5 to 54.5 Gy), well below the ablative doses expected with SBRT.

A safe way to increase the delivered BED_10,_ without jeopardizing the treatment’s safety regarding the critical surrounding gastrointestinal OARs is to resort to an isotoxic dose prescription (IDP). While allowing to protect the OARs with the use of predetermined OARs tolerance levels, the dose delivered to the GTV and TVI can be individually increased to the chosen maximum level [14]. With IDP, the threshold of BED_10_ > 60–70 Gy associated with an improved survival in several studies can be safely reached and, as in the current study, the 70 Gy threshold was crossed (related BED_10_ of the SIB-PTV3 Dmean: 73.8 Gy [IQR 70.5–77.3]) [7,9,10,11,12]. We previously reported promising results regarding the integration of iHD-SBRT into a TNT for localized PDAC in our pilot prospective study including our first 39 BR/LA patients [7]. The whole neoadjuvant sequence with iHD-SBRT was feasible (TNT completed in 87.2% of the cases), displayed a safe toxicity profile (acute and late gastrointestinal grade 3 toxicity rates of 2.9 and 4.2%, respectively) and showed favorable surgical and oncological outcomes (median OS: 24.5 months) [7].

To date, no randomized phase II/III trials are available and only few retrospective non-randomized studies attempted to compare conventional CRT with SBRT, mainly for LA patients and only for low to moderate BED_10_ SBRT. In 2017, the retrospective review of De Geus et al. including 14,331 patients with unresected PDAC was the first to detect a survival advantage of SBRT over conventional CRT (*n* = 322; median regimen: 30 Gy in 3 fractions; 13.9 vs. 11.6 months, *p* = 0.018) and IMRT techniques only (13.9 vs. 12.2 months, *p* = 0.049). However, there was no significant survival benefit from adding SBRT to multi-agent chemotherapy (median OS, 14.8 vs. 12.9 months, *p* = 0.095) [22]. In a meta-analysis of LA patients, the outcomes of 870 patients treated with CRT (mainly 50.4 Gy in 28 fractions) were compared to those of 277 patients treated with SBRT (median regimen: 30 Gy in 5 fractions). A modest survival advantage in favor of SBRT was shown only for the 2-year OS (random effect estimate: 26.9 vs. 13.7%, *p* = 0.004) [23]. More recently, another retrospective cohort including 95 LA patients treated with SBRT (median regimen: 28 Gy in 4 fractions, induction chemotherapy: 40%) and 66 with CRT (median regimen: 54 Gy in 28 fractions, concomitant 5-FU) failed to demonstrate a survival benefit for SBRT, including after propensity score-matched analysis, and even reported worse survival results than CRT (1-year OS rates: 66.7 vs. 80% for CRT, *p* = 0.455) [24]. In our study, all the localized PDAC patients included in the TNT sequence were homogeneously treated regarding the induction chemotherapy and the iHD-SBRT protocol. We were also the first to compare the outcomes of patients treated with high-BED_10_ SBRT to those of conventional CRT. A median OS of 22.5 months was obtained for the iHD-SBRT group, which is promising when compared to the literature available [7,8]. (Table 5) For the CRT group, the median OS of 15.9 months was comparable to the results of previous phase III trials [18,25,26]. There was a statistically significant survival benefit in favour of the iHD-SBRT (median OS: 22.5 vs. 15.9 months, *p* < 0.001; median PFS: 16.7 vs. 11.5 months, *p* = 0.011) and this was also the case for the subgroup of patients who did not have an oncological resection after RT (*n* = 59; median OS: 20.7 vs. 15.9 months; *p* = 0.048). The survival advantage of iHD-SBRT over conventional CRT was still statistically significant on multivariate analysis (HR 0.39 [CI95% 0.27–0.83]). The 1-year LC after iHD-SBRT (75.8%) was comparable to the previous results of the main phase II trials available studying SBRT and was significantly better compared to the CRT group (39.3%) (Table 5).

Our study presents several limitations. First, our study was retrospective and, although the main baseline characteristics of the two groups were well balanced, this cannot replace a true randomized study. Secondly, in the conventional CRT cohort, fewer patients have had a surgery as the surgical vascular reconstruction techniques for pancreatectomy at that time did not allow for it and a majority of patients were treated before the introduction of modern multi-agent chemotherapy (mFFx and Gem-Np). Compared to Gemcitabine alone, multi-agent chemotherapy significantly improved survival in the adjuvant setting as well as for metastatic pancreatic cancer and, therefore, has recently been widely used in the neoadjuvant strategy, as in our SBRT cohort [31,32]. Therefore, although we identified these factors among the main confounding factors for mortality and adjusted our results accordingly, all the biases inherent to retrospective study could not be completely eliminated. Third, the number of patients in each group remained limited as it was a single center analysis and included both BR and LA patients. Therefore, generalization of these results should be done with caution.

In this disease where the prognosis remains somber, more effective systemic therapies are gradually paving the way for RT to be meaningful in survival outcomes. To this end, iHD-SBRT is an attractive treatment option, allowing to be easily integrated into maximized neoadjuvant strategies and further studies are urgently warranted.

## 5. Conclusions

In conclusion, iHD-SBRT is a promising RT option and may offer an improvement in OS in comparison to conventional CRT for localized PDAC. Further validation is required to confirm the exact role of iHD-SBRT and the optimal therapeutic sequence for the treatment of localized PDAC. For this purpose, our group launched the randomized phase II STEREOPAC trial [NCT05083247] aiming to compare mFFX alone versus mFFX + iHD-SBRT as neoadjuvant strategies in 256 patients with BR pancreatic cancer [33].

## Figures and Tables

**Figure 1 cancers-14-05730-f001:**
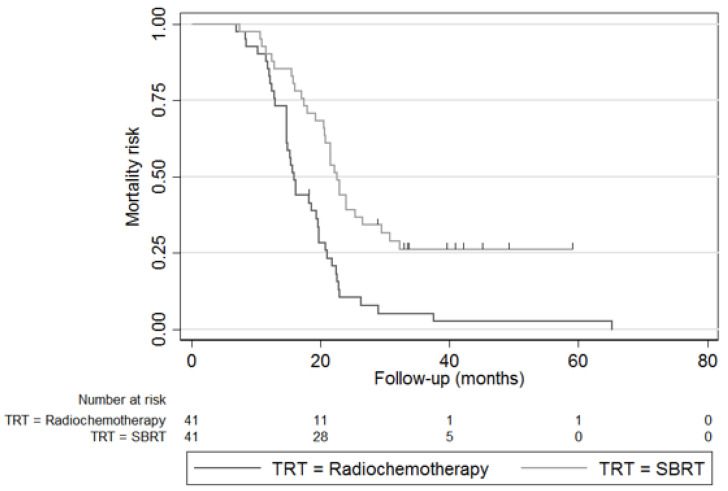
Kaplan–Meier plot of overall survival of the iHD-SBRT and CRT cohort (*n* = 82).

**Figure 2 cancers-14-05730-f002:**
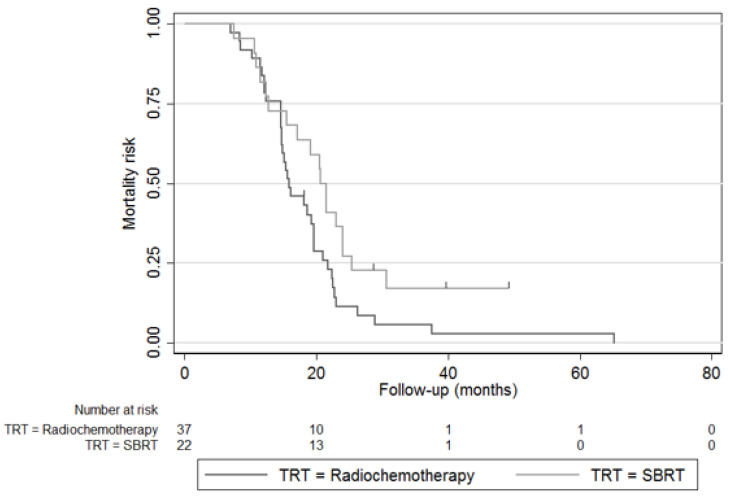
Kaplan–Meier plot of overall survival of the iHD-SBRT and CRT cohort for patients without oncological resection only (*n* = 59).

**Table 1 cancers-14-05730-t001:** Baseline characteristics of the iHD-SBRT and the conventional CRT groups.

	Global Cohort (*n* = 82)	CRT Group (*n* = 41)	iHD-SBRT Group (*n* = 41)	*p*-Value Chi ²
**Gender**				0.264
Female (*n* = 35)	42.7%	48.8%	36.6%	
Male (*n* = 47)	57.3%	51.2%	63.4%	
**Age (years)**				0.269
<60 (*n* = 41)	50.0%	56.1%	43.9%	
≥60 (*n* = 41)	50.0%	43.9%	56.1%	
**CA19.9 values at diagnosis (kU/L)**				0.359
<200 (*n* = 52)	63.4%	58.5%	68.3%	
≥200 (*n* = 30)	36.6%	41.5%	31.7%	
**Primary Site**				0.027
Head/uncus/isthmus (*n* = 59)	72.0%	61.0%	82.9%	
Body/tail (*n* = 23)	28.0%	39.0%	17.1%	
**Tumour diameter (mm)**				0.824
<40 (*n* = 45)	54.9%	56.1%	53.7%	
≥40 (*n* = 37)	45.1%	43.9%	46.3%	
**Staging TNM 8th ed.**				0.422
IB (*n* = 9)	11.0%	12.2%	9.8%	
II a/b (*n* = 19)	23.2%	17.1%	29.3%	
III (*n* = 54)	65.8%	70.7%	60.9%	
**Resection status**				0.656
Borderline (*n* = 36)	43.9%	41.5%	46.3%	
Locally advanced (*n* = 46)	56.1%	58.5%	53.7%	
**Number of induction CT cycles**				<0.001
0–3 (*n* = 22)	26.8%	51.2%	2.4%	
4–8 (*n* = 48)	58.6%	39.0%	78.0%	
>8 (*n* = 12)	14.6%	9.8%	19.5%	
**Time of induction (months)**				0.002
<2 (*n* = 22)	26.8%	43.9%	9.8%	
≥2– < 4 (*n* = 38)	43.6%	39.0%	53.7%	
≥4 (*n* = 22)	26.8%	17.1%	36.5%	
**Type of induction CT**				<0.001
None (*n* = 10)	12.2%	24.4%	0.0%	
mFFX/Gem-Np (*n* = 53)	64.6%	29.3%	100.0%	
Gem-based, other than Gem-Np (*n* = 19)	23.2%	46.3%	0.0%	
**Oncological resection**				<0.001
No (*n* = 59)	72.0%	90.2%	53.7%	
Yes (*n* = 23)	28.0%	9.8%	46.3%	
				***p*-value Wilcoxon Test**
**Age (years), median [IQR]**	60.2 (53.0–67.7)	58.0 (53.0–67.0)	61.5 (54.0–69.6)	0.228
**CA19.9 value at diagnosis (kU/L), median [IQR]**	86.4 (14.3–502.0)	160.0 (21.42–582.5)	60.4 (9.0–210.0)	0.312
**Tumour diameter (mm), median [IQR]**	37.5 (32.0–45.0)	38.0 (33.0–45.0)	37.0 (32.0–44.0)	0.442
**Number of CT cycles (induction), median [IQR]**	6 (3–8)	3 (0–5)	7 (6–8)	<0.001
**Time of induction (months), median [IQR]**	2.8 (1.9–4.2)	2.1 (0.8–3.3)	3.7 (2.6–4.6)	<0.001
**Number of RT fractions, median [IQR]**	14 (5–25)	25 (25–28)	5 (5–5)	<0.001

iHD-SBRT = isotoxic high-dose stereotactic body radiation therapy; CRT = chemoradiotherapy; CT = chemotherapy; mFFX = modified FOLFIRINOX; Gem-Np = gemcitabine/nab-paclitaxel, Gem = gemcitabine; IQR = interquartile range; RT = radiotherapy.

**Table 2 cancers-14-05730-t002:** Treatment plan analysis for the PTVs and related BED_10_.

	iHD-SBRT (*n* = 41)	CRT (*n* = 41)	*p*-Value Wilcoxon Test
**PTV1**			
Median volume, cm^3^ (IQR)	99.6 (77.0–121.9)	422.7 (277.2–691.3)	<0.001
Mean dose (Gy), median (IQR)	37.7 (35.7–39.2)	50.2 (47.8–52.9)	<0.001
Related BED_10_ (Gy), median (IQR)	66.1 (61.2–69.9)	60.3 (57.0–63.4)	<0.001
**SIB-PTV (PTV3)**			
Median volume, cm^3^ (IQR)	71.4 (61.5–94.5)	/	/
Mean dose (Gy), median (IQR)	40.7 (39.4–42.0)	/	/
Related BED_10_ (Gy), median (IQR)	73.8 (70.5–77.3)	/	/
**Dmax**			
Mean Dmax (Gy), median (IQR)	52.0 (49.1–52.5)	56.4 (51.7–61.6)	<0.001
Related BED_10_ (Gy), median (IQR)	106.1 (97.3–107.6)	68.4 (61.3–7.4.6)	<0.001

PTV = planning target volume; IQR = interquartile range; Gy = gray; BED_10_ = biologically effective dose (α/β = 10); Dmax = maximum dose, SIB = simultaneous integrated boost.

**Table 3 cancers-14-05730-t003:** Univariate Cox regression analyses (*n* = 82).

	HR (CI 95%)	*p*-Value
**Gender**		0.611
Female	1	
Male	1.13 (0.70 to 1.83)	
**Age (years)**		0.838
<60	1	
≥60	1.05 (0.65 to 1.69)	
**CA19.9 values at diagnosis (kU/L)**		0.990
<200	1	
≥200	0.99 (0.61 to 1.63)	
**Primary Site**		0.851
Head/uncus/isthmus	1	
Body/tail	1.05 (0.63 to 1.76)	
**Tumour diameter (mm)**		0.776
<40	1	
≥40	0.93 (0.58 to 1.50)	
**Staging TNM 8th ed.**		0.874
IB	1	
II A/B	1.08 (0.46 to 2.52)	
III	0.93 (0.44 to 1.97)	
**Resection status**		0.277
Borderline	1	
Locally advanced	0.77 (0.47 to 1.24)	
**Number of induction CT cycles**		**0.001**
0–3	1	
4–8	0.47 (0.27 to 0.81)	
>8	0.23 (0.09 to 0.54)	
**Time of induction (months)**		**0.005**
<2	1	
≥2 & <4	0.46 (0.26 to 0.81)	
≥4	0.38 (0.20 to 0.72)	
**Type of induction CT**		**0.025**
None	1	
Gem-based (except Gem/Np)	0.69 (0.31 to 1.51)	
mFFX/Gem-Np	0.42 (0.20 to 0.84)	
**Type of Radiotherapy**		**<0.001**
CRT	1	
iHD-SBRT	0.39 (0.24 to 0.64)	
**Oncological resection**		**0.009**
No	1	
Yes	0.47 (0.27 to 0.83)	

iHD-SBRT = isotoxic high-dose stereotactic body radiation therapy; CRT = chemoradiotherapy; CT = chemotherapy; mFFX = modified FOLFIRINOX; Gem-Np = gemcitabine/nab-paclitaxel, Gem = gemcitabine; HR = hazard ratio.

**Table 4 cancers-14-05730-t004:** Multivariate Cox regression analyses (*n* = 82).

Variables	Model 1 HR Adjusted (CI 95%)	*p*-Value	Model 2 HR Adjusted (CI 95%)	*p*-Value
**Type of RT**		**0.007**		**0.014**
CRT	1		1	
iHD-SBRT	0.46 (0.26 to 0.81)		0.39 (0.18 to 0.83)	

Model 1 = model adjusted for oncological resection. Model 2 = model adjusted for oncological resection, type of induction chemotherapy, number of chemotherapy cycles (induction) and time of induction.

**Table 5 cancers-14-05730-t005:** Comparison of our survival outcomes with selected CRT and modern pancreatic SBRT trials available in the literature.

Study	Study Design	Type of RT	*N*	Res. Status	Dose (Gy)/#	Chemotherapy	RR (%)	R0 RR (0 mm, %)	1y-LC (%)	mOS (months)
**Current study**	Retro	iHD-SBRT CRT	41 41	BR (46%) LA BR (41.5%) LA	35–40/5 (SIB TVI up to 53 Gy) 45–60 Gy/25–30	I: mFFX or Gem-Np C: / I: gem-based (46%), mFFx/Gem-Np (29%), none (25%) C: Gem-based or 5FU	46.3 9.8	73.7 33.3	75.8% 39.3%	22.5 15.9
Barhoumi et al., 2013 [25]	Phase III	CRT	Arm B: 59	LA	60/30	I: / C = 5-FU/cisplatin	3	NR	NR	11.1
Herman et al., 2015 [27]	Phase II	SBRT	49	LA	33/5	I: gem C: gem	8	100	78	13.9
Hammel et al., 2016 [18]	Phase III	CRT	Arm D: 133	LA	54/30	I: gem +/− erlotinb C: capecitabine	3	61	NR	15.2
Quan et al., 2017 [28]	Phase II	SBRT	35	BR (54%) LAPC	36/3	I: gem/capecitabine (4 cycles) C: /	33	91.7	70.5	18.3
Jang et al., 2018 * [29]	Phase II/III	CRT	Arm B: 27	BR	54/30	I: / C: Gem	63	82.4		21
Suker et al., 2019 [30]	Phase II	SBRT	50	LA	40/5	I: mFFX	12	100	NR	15
Versteijne et al., 2020 [20]	Phase III	CRT	Arm B: 119	R, BR	36/15	I: / C: gem	61	71		16
Simoni et al., 2021 [14]	Observational	SBRT	59	BR (46%) LA	25–30/5 (SIB TVI 50 Gy)	I: mFFX or Gem-Np (6 to 12 cycles). C: /	59.4	57.1	79.7	30.2
Fietkau et al., 2022 (Abstract) [19]	Phase III	CRT	Arm B: 168	LA	50.4/28	I: Gem C: Gem	36.3	25	NR	2-Yr OS: 34.8%
Katz et al., 2022 * [8]	Random. Phase II	SBRT	Arm B: 40	BR	25–33/5 (SIB TVI up to 40 Gy)	I: mFFX (7 cycles) C: /	51	33	NR	17.1

SBRT: stereotactic body radiation therapy; iHD-SBRT: isotoxic high dose stereotactic body radiation therapy; CRT: chemoradiotherapy; Res.: resection; RR: resection rate; mOS: median overall survival; Retro: retrospective; Random.: randomized; N: number of patients; R: resectable; BR: borderline resectable; LA: locally advanced; Gy: Gray; #: number of fractions; SIB: simultaneous integrated boost; TVI: tumour vessel interface; mFFX: modified FOLFIRINOX; I: induction; C: concomitant; Gem: gemcitabine; Gem-Np: gemcitabine/nab-paclitaxel; 1y-LC: 1 year local control; NR: not reported; *: terminated after interim analysis.

## Data Availability

Additional data can be obtained through direct request to the corresponding author within the limit of what can be disclosed to respect the data protection related to ethical recommendations.

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
