# Peer review of "Isotoxic High-Dose Stereotactic Body Radiotherapy (iHD-SBRT) Versus Conventional Chemoradiotherapy for Localized Pancreatic Cancer: A Single Cancer Center Evaluation"

_cancers, 2022, doi:10.3390/cancers14235730_

Round 1
Reviewer 1 Report
This is a very well written interesting paper on a comparison between stereotactic body radiotherapy and conventional chemotherapy, in patients.The results are promising and very well presented.
I only have very minor comments/suggestions:
1)It might be good to consider adding the patient characteristics in the methodology section rather than the results, and then cross-reference them in the results.
2)The tables are informative and so are the graphs. It might be useful (especially for non clinical readers) to add a sketch/illustration on your overall methodology and procedure (e.g. a treatment chronological diagram/schematic, like a graphical abstract). This would be useful.
3) I agree with you that results need to be taken with caution as this is only one study, however, I suggest you do not end the discussion with this paragraph. Maybe include an additional short paragraph of a few sentences highlighting again the importance/impact of your work (in other words end with a more positive note as this is a very good study).
The above are minor points. I think the paper is almost ready for publication
Author Response
Reviewer(s)' Comments to Author:
Reviewer: 1
This is a very well written interesting paper on a comparison between stereotactic body radiotherapy and conventional chemotherapy, in patients.The results are promising and very well presented.
Response: We would like to thank the reviewer for the positive and constructive comments that are very helpful to establish a more comprehensive and clearer manuscript.
I only have very minor comments/suggestions:
- It might be good to consider adding the patient characteristics in the methodology section rather than the results, and then cross-reference them in the results.
Response: patient characteristics and Table 1 have been added in the methodology section. A cross-reference has been created in the Results section.
- The tables are informative and so are the graphs. It might be useful (especially for non clinical readers) to add a sketch/illustration on your overall methodology and procedure (e.g. a treatment chronological diagram/schematic, like a graphical abstract). This would be useful.
Response: a graphical abstract has been added.
- I agree with you that results need to be taken with caution as this is only one study, however, I suggest you do not end the discussion with this paragraph. Maybe include an additional short paragraph of a few sentences highlighting again the importance/impact of your work (in other words end with a more positive note as this is a very good study).
Response: this paragraph has been added to the end of the discussion:
“In this disease where the prognosis remains somber, more effective systemic therapies are gradually paving the way for RT to be meaningful in survival outcomes. To this end, iHD-SBRT is an attractive treatment option, allowing to be easily integrated into maximized neoadjuvant strategies and further studies are urgently warranted.”
Reviewer 2 Report
A thorough analysis of your experience, obviously retrospective in nature. Some remarks:
1) Major limitation is that your iHD SBRT cohort has had standard modern multi agent chemotherapy whereas in your conventional CRT cohort only a minority had this. This weakens the conclusion although you try to compensate for this in your multivariate analysis. Despite the latter, you cannot rule out that the type of chemotherapy intervenes with the outcomes of both cohorts. Perhaps you should stress this more in your discussion .
2) Throughout the paper you write about median survival as main endpoint for outcome. I agree that this is often used by authors. Yet for studies of neoadjuvant or "induction" therapy median survival is a poor endpoint since it is in the steep part of the curves and thus can be influenced by all kind of biases, in particular selection and time biases. Longterm outcomes 3 or 5 y OS is much more robust.
3) very minor: In the last sentence you mention your STEREOPAC trial (I'm looking forward to its results) in which you are coing to compare FFX with or without SBRT ".....256 BR patients [27]." To resect means: to remove/ take out. Even pancreatic cancer surgeons cannot resect patients, only tumours.
Author Response
Reviewer(s)' Comments to Author:
Reviewer2
First, we would like to thank the reviewer for the constructive comments that are very helpful to establish a more comprehensive and clearer manuscript.
A thorough analysis of your experience, obviously retrospective in nature.
Some remarks:
- Major limitation is that your iHD SBRT cohort has had standard modern multi agent chemotherapy whereas in your conventional CRT cohort only a minority had this. This weakens the conclusion although you try to compensate for this in your multivariate analysis. Despite the latter, you cannot rule out that the type of chemotherapy intervenes with the outcomes of both cohorts. Perhaps you should stress this more in your discussion .
Response: the following paragraph has been modified in the discussion:
“Our study presents several limitations. First, our study was retrospective and, although the main baseline characteristics of the two groups were well balanced, this cannot not replace a true randomized study. Secondly, in the conventional CRT cohort, fewer patients have had a surgery as the surgical vascular reconstruction techniques for pancreatectomy at that time did not allow for it and a majority of patients were treated before the introduction of modern multi-agent chemotherapy (mFFx and Gem-Np). Compared to Gemcitabine alone, multi-agent chemotherapy significantly improved survival in the adjuvant setting as well as for metastatic pancreatic cancer and, therefore, has recently been widely used in the neoadjuvant strategy, as in our SBRT cohort. [27-28] So, although we identified these factors among the main confounding factors for mortality and adjusted our results accordingly, all the biases inherent to retrospective study could not be completely eliminated. Third, the number of patients in each group remained limited as it was a single center analysis and included both BR and LA patients. Therefore, generalization of these results should be done with caution.”
The following references have been added in the manuscript:
Conroy T, Desseigne F, Ychou M, Bouche O, Guimbaud R, Becouarn Y, Adenis A, Raoul JL, Gourgou-Bourgade S, de la Fouchardière C et al. FOLFIRINOX versus gemcitabine for metastatic pancreatic cancer. N Engl J Med. 2011;364(19):1817–25.
Conroy T, Hammel P, Hebbar M, Ben Abdelghani M, Wei AC, Raoul JL, Choné L, François E, Artu P, Biagi JJ et al. FOLFIRINOX or gemcitabine as adjuvant therapy for pancreatic cancer. N Engl J Med. 2018; 379:2395-406.
- Throughout the paper you write about median survival as main endpoint for outcome. I agree that this is often used by authors. Yet for studies of neoadjuvant or "induction" therapy median survival is a poor endpoint since it is in the steep part of the curves and thus can be influenced by all kind of biases, in particular selection and time biases. Longterm outcomes 3 or 5 y OS is much more robust.
Response: we agree with the reviewer that these long-term survival results are indeed important outcomes. Unfortunately, as the long-term follow-up of our iHD-SBRT cohort is still limited, the 5-y OS rates were not yet available. For the 3-y OS rate, the results were still partial at the time of the analysis and therefore, due to their limited statistical value, we chose not to add these partial data in the manuscript. The SBRT long-term outcomes will be thoroughly analysed when a sufficient follow-up time will be reached. Meanwhile, we added in the Results chapter of the manuscript the 2-y OS rates, which were available for the whole cohort (10.0% vs 43.9%; p= .001 in favour of the SBRT cohort).
- very minor: In the last sentence you mention your STEREOPAC trial (I'm looking forward to its results) in which you are coing to compare FFX with or without SBRT ".....256 BR patients [27]." To resect means: to remove/ take out. Even pancreatic cancer surgeons cannot resect patients, only tumours.
Response: the following sentence has been modified in the conclusion:
“For this purpose, our group launched the randomized phase II STEREOPAC trial [NCT05083247] aiming to compare mFFX alone versus mFFX + iHD-SBRT as neoadjuvant strategies in 256 patients with BR pancreatic cancer [28].”
And we thank the reviewer for the interest in the STEREOPAC trial which will soon open for inclusion.